# On computation of Hough functions

Houjun Wang[1,2], John P. Boyd[3], and Rashid A. Akmaev[2]

[1]CIRES, University of Colorado Boulder, Boulder, Colorado, USA.
[2]Space Weather Prediction Center, NOAA, Boulder, Colorado, USA.
[3]Department of Climate and Space Sciences and Engineering, University of Michigan, Ann Arbor, Michigan, USA.

*Correspondence to:* Houjun Wang (houjun.wang@noaa.gov)

**Abstract.**

Hough functions are the eigenfunctions of the Laplace's tidal equation governing fluid motion on a rotating sphere with a resting basic state. Several numerical methods have been used in the past. In this paper, we compare two of those methods: *normalized* associated Legendre polynomial expansion and Chebyshev collocation. Both methods are not widely used, but both have some advantages over the commonly-used unnormalized associated Legendre polynomial expansion method. Comparable results are obtained using both methods. For the first method we note some details on numerical implementation. The Chebyshev collocation method was first used for the Laplace tidal problem by Boyd (1976) and is relatively easy to use. A compact MATLAB code is provided for this method. We also illustrate the importance and effect of including a *parity factor* in Chebyshev polynomial expansions for modes with *odd* zonal wavenumbers.

## 1  Introduction

Hough functions are the eigenfunctions of the eigenvalue problem of the following form:

$$\mathcal{F}(\Theta) + \gamma\Theta = 0, \tag{1}$$

where $\mathcal{F}$ is a linear differential operator, the *Laplace's tidal operator*, defined as:

$$\mathcal{F}(\Theta) \equiv \frac{d}{d\mu}\left(\frac{1-\mu^2}{\sigma^2-\mu^2}\frac{d\Theta}{d\mu}\right)$$
$$-\frac{1}{\sigma^2-\mu^2}\left[\frac{s}{\sigma}\frac{\sigma^2+\mu^2}{\sigma^2-\mu^2}+\frac{s^2}{1-\mu^2}\right]\Theta, \tag{2}$$

with $\mu = \sin\phi \in [-1,1]$, $\phi$ the latitude, $s$ the zonal wavenumber, and $\sigma$ the dimensionless frequency normalized by $2\Omega$ ($\Omega$ the earth's rotation rate), while

$$\gamma \equiv \frac{4a^2\Omega^2}{gh} \tag{3}$$

is the Lamb's parameter (Andrews et al., 1987, p. 154), with $a$ the earth's radius, $g$ the acceleration due to the earth's gravity, and $h$ the so-called *equivalent depth*.

Several numerical methods have been used to solve the eigenvalue problem for the Laplace tidal equation in the past. Hough (1898) pioneered the solutions of the Laplace tidal equations using spherical harmonic expansion, or equivalently *spherical harmonic Galerkin* method, so eigenfunctions of the eigenvalue problem Eq. (1) that describe the latitudinal dependence are often called *Hough functions* (Flattery, 1967; Longuet-Higgins, 1968; Lindzen and Chapman, 1969). Each function of latitude and longitude is expanded as a Fourier series in longitude using the usual Fourier functions, $\cos(s\lambda)$ and $\sin(s\lambda)$, where $s$, an integer, is the "zonal wavenumber", $\lambda$ is the longitude. Each longitudinal trigonometric function is multiplied by a latitudinal basis function which depends on the zonal wavenumber $s$. Hough and his successors used a latitudinal basis of *unnormalized* associated Legendre polynomials (ALPs). Both Kato (1966) and Flattery (1967) used the *method of continued fractions* to solve for eigenvalues one by one with iterations. This is not the most convenient method to work with and some eigenvalues could be missed. Chen and Lu (2009) also discussed calculation of Hough functions following the same original formulation without showing any details on numerical procedures.

Computation of Hough functions based on expansion in terms of *normalized* ALPs was first used by Dikii (1965). It was later elaborated in a note by Groves (1981), along with a method of evaluating related wind functions. Jones (1970) used group-theoretical methods to obtain a matrix representation of Hough functions by expanding in normalized spherical harmonics.

Although it is closely related to the original method of expansion in terms of *unnormalized* ALPs, expansion in terms of the *normalized* ALPs leads to two symmetric matrices for symmetric and anti-symmetric modes. This has both *computational and conceptual* advantages over the original expansion in unnormalized ALPs: 1) the eigenvalue problem of symmetric matrix can be solved very accurately by Jacobi method (e.g., Demmel and Veselić, 1992), and 2) symmetry guarantees that all of the "eigenvalues are real and that there is an orthonormal basis of eigenvectors" (Golub and Van Loan, 1996, p. 393).

There is also another way of computing Hough functions or *global normal modes*, such as Longuet-Higgins (1968); Kasahara (1976); Žagar et al. (2015), also using spherical harmonic expansion, in which the equivalent depth is assigned (for each zonal wavenumber) and the frequency of the normal modes are obtained as the eigenvalues. This is different from eigenvalue problem for tidal waves in which the wave frequencies and zonal wavenumber are specified and eigenvalues are obtained and used to compute equivalent depths, just as stated in the original eigenvalue problem Eq. (1).

The collocation method was first applied to compute Hough functions by Boyd (1976). His latitudinal basis functions replace associated Legendre functions by cosine functions of colatitude $\varphi$ multiplied by a "parity factor" which is $\sin(\varphi)$ for *odd* zonal wavenumber $s$ and the constant one for even zonal wavenumbers. The parity factor is explained in Appendix C. In addition, the modified latitudinal variable

$$\mu \equiv \cos(\varphi) = \sin(\phi) \in [-1, 1]$$

is often used to analyze and solve differential equations in spherical geometry. The reason is that trigonometric functions are replaced by powers of $\mu$, simplifying almost everything. And denoting the Chebyshev polynomials by $T_n(x)$, Chebyshev's famous identity shows that

$$T_n(\mu) = T_n(\cos(\varphi)) = \cos(n\varphi), \quad n = 0, 1, \ldots.$$

Thus a Fourier cosine series in colatitude is, with the same coefficients, also a Chebyshev polynomial series in $\mu$.

Boyd (1976) and Orszag (1974) listed several advantages of Chebyshev polynomial collocation over spherical harmonic Galerkin approximations. First, cosines/Chebyshev polynomials are much simpler than associated Legendre functions, which are different for each different zonal wavenumber $s$. Second, collocation, which *evaluates* and *interpolates*, is much easier to program than the Galerkin method, which *integrates*. These advantages make it much easier to apply the Chebyshev collocation method than the spherical harmonic Galerkin method. See also (Hesthaven et al., 2007, Chapter 3) for a discussion of advantages of Fourier-collocation methods over the Fourier-Galerkin methods.

In this paper we compare the solution of the eigenvalue problem for the Laplace tidal operator using two numerical methods, the *normalized* ALP expansion method and the Chebyshev collocation method. Both methods are not widely used, but both have some advantages over the commonly-used unnormalized ALP expansion. For the first method we note some details of numerical implementation as the denominators in some terms of matrix entries can become zero. For the second method a compact MATLAB code is provided to facilitate its use. We also discuss other related issues and show that there is no accuracy penalty in using the Chebyshev collocation method.

## 2    Computation of Hough functions

In this section, we compare two methods for computing Hough functions: one using the *normalized* associated Legendre polynomial (ALP) expansion, the other using the Chebyshev collocation method.

### 2.1    Computation of Hough functions using normalized associated Legendre polynomial expansion

The first method uses the expansion in terms of *normalized* associated Legendre polynomials (ALPs) (e.g., Groves, 1981). To solve the Laplace's tidal equation, first expand $\Theta$ in terms of the *unnormalized* associated Legendre polynomials $P_r^s$

$$\Theta = \sum_{r=s}^{\infty} c_r P_r^s(\mu). \tag{4}$$

Substituting into the Laplace tidal equation Eq. (1), one obtains

$$Q_{r-2}c_{r-2} + (M_r - \lambda)c_r + S_{r+2}c_{r+2} = 0, \qquad (r \geqslant s), \tag{5}$$

where

$$Q_{r-2} = \frac{(r-s)(r-s-1)}{(2r-1)(2r-3)[s/\sigma - r(r-1)]}, \tag{6a}$$

$$M_r = \frac{\sigma^2[r(r+1) - s/\sigma]}{r^2(r+1)^2}$$

$$+ \frac{(r+2)^2(r+s+1)(r-s+1)}{(r+1)^2(2r+3)(2r+1)[s/\sigma - (r+1)(r+2)]}$$

$$+ \frac{(r-1)^2(r^2-s^2)}{r^2(4r^2-1)[s/\sigma - r(r-1)]}, \tag{6b}$$

$$S_{r+2} = \frac{(r+s+2)(r+s+1)}{(2r+3)(2r+5)[s/\sigma - (r+1)(r+2)]}, \tag{6c}$$

and

$$\lambda = \frac{gh}{4a^2\Omega^2} = \frac{1}{\gamma}. \tag{7}$$

These equations were first given by Hough (1898); see also Lindzen and Chapman (1969).

The *normalized* associated Legendre polynomials $P_{r,s}$ are defined in terms of the *unnormalized* associated Legendre polynomials $P_r^s$ by

$$P_{r,s} = \left[\frac{2(r+s)!}{(2r+1)(r-s)!}\right]^{-\frac{1}{2}} P_r^s. \tag{8}$$

Expanding $\Theta$ in terms of the *normalized* associated Legendre polynomials $P_{r,s}$

$$\Theta = \sum_{r=s}^{\infty} a_r P_{r,s}(\mu), \tag{9}$$

we have (Dikii, 1965; Groves, 1981)

$$L_{r-2}a_{r-2} + (M_r - \lambda)a_r + L_r a_{r+2} = 0 \qquad (r \geqslant s), \tag{10}$$

where

$$L_r = \frac{[(r+s+1)(r+s+2)(r-s+1)(r-s+2)]^{\frac{1}{2}}}{(2r+3)[(2r+2)(2r+5)]^{\frac{1}{2}}[s/\sigma - (r+1)(r+2)]}, \tag{11a}$$

$$M_r = -\frac{\sigma^2 - 1}{(s/\sigma + r)(s/\sigma - r - 1)}$$
$$+ \frac{(r-s)(r+s)(s/\sigma - r + 1)}{(2r-1)(2r+1)(s/\sigma + r)[s/\sigma - r(r-1)]}$$
$$+ \frac{(r-s+1)(r+s+1)(s/\sigma + r + 2)}{(2r+1)(2r+3)(s/\sigma - r - 1)[s/\sigma - (r+1)(r+2)]}. \tag{11b}$$

Equation (10) can be written in a matrix form for the coefficients vector $x = [a_s, a_{s+1}, a_{s+2}, a_{s+3}, \ldots]^T$ as the matrix eigenvalue problem $F_0 x = \lambda x$, with matrix $F_0$ defined as

$$F_0 = \begin{bmatrix}
M_s & 0 & L_s & 0 & 0 & \cdots \\
0 & M_{s+1} & 0 & L_{s+1} & 0 & \cdots \\
L_s & 0 & M_{s+2} & 0 & L_{s+2} & \cdots \\
0 & L_{s+1} & 0 & M_{s+3} & 0 & \cdots \\
0 & 0 & L_{s+2} & 0 & M_{s+4} & \cdots \\
\vdots & \vdots & \vdots & \vdots & \vdots & \ddots
\end{bmatrix}. \tag{12}$$

Or it may be written as, respectively, $F_1 x_1 = \lambda_1 x_1$, $x_1 = [a_s, a_{s+2}, \ldots]^T$ for symmetric modes, with matrix $F_1$ defined as

$$F_1 = \begin{bmatrix}
M_s & L_s & 0 & 0 & \cdots \\
L_s & M_{s+2} & L_{s+2} & 0 & \cdots \\
0 & L_{s+2} & M_{s+4} & L_{s+4} & \cdots \\
\vdots & \vdots & \vdots & \vdots & \ddots
\end{bmatrix}, \tag{13}$$

and $F_2 x_2 = \lambda_2 x_2$, $x_2 = [a_{s+1}, a_{s+3}, \ldots]^T$ for antisymmetric modes, with matrix $F_2$ defined as

$$
F_2 = \begin{bmatrix}
M_{s+1} & L_{s+1} & 0 & 0 & \ldots \\
L_{s+1} & M_{s+3} & L_{s+3} & 0 & \ldots \\
0 & L_{s+3} & M_{s+5} & L_{s+5} & \ldots \\
\vdots & \vdots & \vdots & \vdots & \ddots
\end{bmatrix}. \tag{14}
$$

These are real symmetric matrices and the eigenvalue problem can be solved accurately using the Jacobi methods (e.g., Golub and Van Loan, 1996, Chapter 8). The computed eigenvectors are the expansion coefficients.

A few remarks on unnormalized versus normalized ALP expansion are in order here. The unnormalized polynomials (not just ALPs, but Legendre and Chebyshev and Hermite polynomials too) have survived because the canonical unnormalized forms have polynomial coefficients that are integers or rational numbers. This is convenient for many applications, such as when using exact arithmetic in computer algebra. Note that this property carries over to the Galerkin matrix elements for the Hough differential equation, which are rational functions of $r$ and $s$ in Eq. (6). Also, for some purposes it is very convenient

to use polynomials which are all 1 at $\mu = 1$, as true for unnormalized Chebyshev and Legendre polynomials. The bad news is that unnormalized polynomials generate bigger roundoff errors in all calculations, not just computing matrix eigenvalues. The Galerkin matrix element formulas are more complicated for normalized polynomials. As we noted above, a particular advantage of working with normalized ALPs is that the discretization matrix becomes a symmetric matrix. Spectral discretizations often generate a few inaccurate eigenvalues with nonzero imaginary parts, but the eigenvalues of a symmetric tridiagonal matrix are

always real.

    A note on numerical implementation is relevant here, since denominators of terms in $M_r$ can become zero. We found that form (6b), instead of (11b), of $M_r$ should be used, even though the two forms are *equivalent*. In addition, we should set that last term of (6b) of $M_r$ to zero when it becomes a form of $0/0$. Thus, to compute the $(s = 2, \sigma = 1)$ modes or SW2 (*semidiurnal, westward propagating, zonal wave number 2*) modes, we should set the last term of (6b) to zero when $r = s = 2$.

The Fortran 90 source code of the Jacobi eigenvalue algorithm implemented by Burkardt (2013) can be used to solve the two symmetric matrix eigenvalue problems. It can actually, for the $(s = 1, \sigma = 0.5)$ modes or DW1 (*diurnal, westward propagating, zonal wave number 1*) tide, compute the one *infinite* eigenvalue with $P_{2,1}$ as the eigenmode, "the most important odd mode" (Lindzen and Chapman, 1969, p. 151) since $P_{2,1} \propto \sin\phi\cos\phi$. So in this way we will not miss any important eigenvalue or eigenfunction; see Section 3 for a discussion on the "missing" modes for the solar diurnal modes and the completeness of

Hough functions. When using MATLAB, we can set any *inf* matrix entry to *realmax* and then use the MATLAB function *eig* to solve the matrix eigenvalue problem. It is also *preferable* to compute eigenvalues for symmetric and anti-symmetric modes separately, especially when there are interior singularities, e.g., for the DW1 tide. A MATLAB implementation is shown in Appendix B1.

    Using the method of expansions in the normalized associated Legendre polynomials, truncated at $r_{max} = 60$ on 94 Gaussian

quadrature points, we compute eigenvalues and eigenfunctions for several important solar tides. We use *solar day* instead of *sidereal day* in our computations. The first several equatorial symmetric and anti-symmetric modes for DW1 are shown in

Fig. 1. The first several equatorial symmetric and anti-symmetric modes for SW2 of scalar fields are shown in Fig. 2(a)-(b). The first several equatorial symmetric and anti-symmetric modes for $(s = 3, \sigma = 1.5)$ modes or TW3 (*terdiurnal, westward propagating, zonal wave number 3*) for temperature field are shown in Fig. 3. For completeness, a method of computing Hough functions for the horizontal wind components by Groves (1981) (with correction) is presented in Appendix A.

## 2.2 Computation of Hough functions using Chebyshev collocation method

The Chebyshev collocation method was first used Boyd (1976) to solve the Laplace tidal problem. Expand $\Theta$ in terms of the Chebyshev polynomials $T_n(\mu)$:

$$\Theta(\mu) = \sin^m \varphi \sum_{n=0}^{N} b_n T_n(\mu), \quad \text{with } m = mod(s, 2), \tag{15}$$

which includes a *parity factor* $\sin \varphi$ for the *odd* zonal wavenumber $s$ (Orszag, 1974; Boyd, 1978), where $\varphi$ is colatitude, $\varphi = \pi/2 - \phi$. See Appendix C for an explanation for parity factor. The Chebyshev collocation points can be defined in different ways. When the interior or "roots" points are used, they are defined as (e.g., Boyd, 2001, p. 571):

$$\mu_i = \cos\left(\frac{(2i-1)\pi}{2N}\right), \quad i = 1, ..., N, \tag{16}$$

where $N$ is total number of collocation points. By using the differential matrices, it is straightforward to apply the Chebyshev collocation methods to any differential operators. Discussion on property of Chebyshev polynomials and collocation method can be found in Boyd (2001) and Trefethen (2000). A MATLAB implementation is shown in Appendix B2.

Parity requirement is discussed in Orszag (1974). To quote from Orszag (1974) "If parity requirements are violated, then differentiability is lost (at the boundaries, i.e., at the poles), possibly resulting in slow convergence of series expansions and associated Gibbs' phenomena. It is important that assumed spectral representations not impose an incorrect symmetry on a solution if infinite-order accurate results are desired" (see also Boyd (1978)).

To show how accuracy is affected by the parity factor, we compare the eigenfunction expansion coefficients $b_n$ computed with or without parity factor in Fig. 4. For both terdiurnal and pentadiurnal tides, when the parity factor is removed, only limited lower-order algebraic convergence rates are achieved: $4^{th}$-order for terdiurnal and $7^{th}$-order for pentadiurnal. When the parity factor is included, spectral or exponential convergence is restored. Thus including the parity factor improves the accuracy dramatically, so solutions are less affected by singularities when they exist. It is important to include the parity factor when computing eigenvalues and eigenfunctions for DW1 $(s = 1, \sigma = 0.5)$ modes (see section 2.3). A theoretical justification for the parity factor is given in Appendix C.

The MATLAB code listed in Appendix B2 includes a *parity factor* for the odd zonal wavenumber. It also computes Hough modes for horizontal wind components. The computed eigenvalue in this case is just (negative) $\gamma$ and from Eq (3) we can compute the corresponding equivalent depths $h$. Hough functions are simply the computed eigenvectors, with different normalization factors that are irrelevant, when Chebyshev differential matrices are used. So the eigenvalue and eigenvector problem we solve can be viewed as a direct discretization of the original operator eigenvalue problem (1).

## 2.3   Comparison of the two methods

Table 1 compares the number of good eigenvalues that can be obtained using the two methods. The "good" eigenvalue is defined as one whose *relative error*

$$E_{\mathrm{rel}}(\hat{\lambda}) = \frac{|\lambda - \hat{\lambda}|}{|\lambda|}$$

is less than $10^{-6}$, where $\lambda$ is the eigenvalue computed at high truncation $N = 160$. This definition is somewhat arbitrary, but is useful for comparisons. It shows that for DW1 about 60% of the computed eigenvalues are good using the normalized ALP expansion method and about 50% of the computed eigenvalues are good using the Chebyshev collocation method; for SW2

a little over 50% of the computed eigenvalues are good using both methods; and for TW3 the number of good eigenvalues is about 75% for both methods. We note that for DW1 only about 15% of the computed eigenvalues are good *without parity factor*, contrasted to 50% *with parity factor*. This again illustrates the importance of preserving correct parity.

Considering the "unusual difficulties" in solving the eigenvalue problem of the Laplace tidal equation using general numerical methods, as remarked by Bailey et al. (1991), it is *remarkable* that Chebyshev collocation method with a parity factor for

odd zonal wavenumber can be used so successfully in solving the eigenvalue problem of the Laplace tidal equation.

## 3   A remark on the completeness of Hough functions

Although the completeness of Hough functions for zonal wavenumber $s$ and period $T = (s+1)/2$ days was questioned earlier by Lindzen (1965), completeness was later proved by Holl (1970) with further analysis by Homer (1992). Giwa (1974) proved by direct computation that, for zonal wavenumber $s$ and period $T = (s+1)/2$ days, Hough functions for tidal oscillations are

the same as the associated Legendre polynomials $P_{s+1}^{s}$ and Hough functions form a *complete* set of orthogonal functions.

One advantage in using the *normalized* associated Legendre polynomials as basis functions, as shown in Section 2.1, is that the eigenvalue problem becomes an eigenvalue problem for two real symmetric matrices, one for symmetric modes and one for anti-symmetric modes. The spectral theory of (Hermitian) symmetric matrices tells us that these real symmetric matrices have "a complete set of orthogonal eigenvectors, and that the corresponding eigenvalues are real" (e.g., Lax, 2002, Chapter 28).

Thus this approach in a heuristic way shows the completeness of Hough functions.

## 4   Summary and Conclusions

In this paper, we briefly survey the numerical methods for computing eigenvalues and eigenvectors for the Laplace tidal operator. In particular we compare two numerical methods: the *normalized* associated Legendre polynomial (ALP) expansion and Chebyshev collocation. The *normalized* ALP expansion method leads to two symmetric matrices which can be solved

very accurately. It also has an advantage in providing another conceptual understanding for the completeness of eigenfunctions (Hough functions) of the Laplace tidal operator. We also note some details on numerical implementation and provide a MATLAB code.

The Chebyshev collocation method was first used by Boyd (1976) for computing the eigenvalues for the Laplace tidal problem. Here we compare this method with the ALP expansion and found that both are producing comparable results. Chebyshev collocation method uses Fourier cosine series in colatitude as the basis functions and is relatively easy to work with. A compact MATLAB code is provided to facilitate the use of Chebyshev collocation method for the Laplace tidal problem.

The Chebyshev polynomial expansion method is merely a Fourier cosine expansion method in disguise (Boyd, 2001). In using the Chebyshev collocation method, it is important to include a *parity factor* in Chebyshev polynomial expansion for *odd* zonal wavenumber modes.

**Appendix A: Hough functions for the horizontal wind components**

Hough function for the horizontal wind components are (Groves, 1981; Lindzen and Chapman, 1969):

$$\Theta_u = \frac{(1-\mu^2)^{\frac{1}{2}}}{\sigma^2 - \mu^2} \left[ \frac{s}{1-\mu^2} - \frac{\mu}{\sigma} \frac{d}{d\mu} \right] \Theta, \tag{A1a}$$

$$\Theta_v = \frac{(1-\mu^2)^{\frac{1}{2}}}{\sigma^2 - \mu^2} \left[ \frac{(s/\sigma)\mu}{1-\mu^2} - \frac{d}{d\mu} \right] \Theta, \tag{A1b}$$

for the eastward and northward components respectively. These can be evaluated numerically by discretizing the differential operators; or evaluated recursively as follows (Groves, 1981). Let

$$S_u = \cos\phi \, \Theta_u, \qquad S_v = \cos\phi \, \Theta_v, \tag{A2}$$

then from Eqs. (A1) we have

$$\sigma S_u - \mu S_v - (s/\sigma)\Theta = 0, \tag{A3a}$$

$$\mu S_u - \sigma S_v - (1/\sigma)\mathcal{D}\Theta = 0, \tag{A3b}$$

where $\mathcal{D} = (1-\mu^2)d/d\mu$. Note that there misses the factor of $1/\sigma$ before $\mathcal{D}\Theta$ in Eq. (40) of Groves (1981). For $s \geqslant 0$, we expand $S_u$ and $S_v$ in terms of the normalized associated Legendre polynomials:

$$S_u = \sum_{r=s}^{\infty} u_r P_{r,s}(\mu), \qquad S_v = \sum_{r=s}^{\infty} v_r P_{r,s}(\mu), \tag{A4}$$

and use Eq. (9) for expansions of $\Theta$, as well as the recurrence relations for the normalized associated Legendre functions (which can be verified or derived from the recurrence relations for the unnormalized associated Legendre polynomials)

$$\mu P_{r,s} = b_r P_{r-1,s} + b_{r+1} P_{r+1,s}, \tag{A5a}$$

$$\mathcal{D}P_{r,s} = (r+1)b_r P_{r-1,s} - r b_r P_{r+1,s}, \tag{A5b}$$

where

$$b_r = [(r^2 - s^2)/(4r^2 - 1)]^{\frac{1}{2}}, \tag{A6}$$

then the coefficients of $P_{r-1,s}$ give

$$b_r u_r = \sigma v_{r-1} - b_{r-1} u_{r-2}$$
$$- (1/\sigma)[(r-2)a_{r-2}b_{r-1} - (r+1)a_r b_r], \tag{A7a}$$
$$b_r v_r = \sigma u_{r-1} - b_{r-1} v_{r-2} - (s/\sigma)a_{r-1}. \tag{A7b}$$

The first several equatorial symmetric and anti-symmetric modes for SW2 ($s = 2, \sigma = 1$) for the zonal wind components computed using the above method are shown in Fig. 2(c)-(f). We also used the second-order central finite difference method to discretize the differential operators in Eqs. (A1a) and (A1b). Comparison of Hough mode computations for wind components using the method presented above and the finite difference method showing no visual differences, except at the two end points where the one-sided finite difference has to be used. The MATLAB code listed in Appendix B1 also computes Hough functions

for the horizontal wind components using the central difference method.

### Appendix B:  Listing of the MATLAB codes for computing Hough functions

In this Appendix, we list the MATLAB codes that can be used to compute eigenvalue and eigenvectors or Hough functions for the Laplace tidal equation. One uses the normalized ALP method and the other uses the Chebyshev collocation method.

### B1    The normalized ALP method

The first MATLAB code uses the normalized ALP method. MATLAB function *pmn_polynomial_value.m* (https://people.sc.fsu.edu/~jburkardt/m_src/legendre_polynomial/pmn_polynomial_value.m) is used to compute normalized associated Legendre polynomials. MATLAB function *lgwt.m* (http://www.mathworks.com/matlabcentral/fileexchange/4540-legendre-gauss-quadrature-weights-and-nodes/content/lgwt.m) is used to compute the Gauss quadrature points. And considering the cumbersome programming with the normalized ALP method, in computing the Hough functions for horizontal

wind components, we use the central difference method with MATLAB function *central_diff.m* (http://www.mathworks.com/matlabcentral/fileexchange/12-central-diff-m/content/central_diff.m).

```
% NALP_HOUGH - Compute Hough functions
% using normalized associated Legendre
% polynomials (ALP)
clear; format long e
a = 6.370d6; g = 9.81d0;
omega = 2.d0*pi/(24.d0*3600.d0);
%s = 1.d0; sigma = 0.4986348375d0; % DW1
 s = 1.d0; sigma = 0.5d0;    % DW1
%s = 2.d0; sigma = 1.0d0;    % SW2
%s = 3.d0; sigma = 1.5d0;    % TW3
N = 62; N2 = N/2; sf = s/sigma;
```

```matlab
    % define L(r) and M(r)
    L = zeros(N,1); M = zeros(N,1);
    for r = s:N+s-1
    i = r-s+1;
    % define L(r)
    L(i) = sqrt((r+s+1)*(r+s+2)*(r-s+1)*(r-s+2))...
            /((2*r+3)*sqrt((2*r+1)*(2*r+5))...
            *(sf-(r+1)*(r+2)));
    % define M(r)
    if (s == 2) && (r == 2)
        M(i) = -(sigma^2*(sf-r*(r+1)))...
                /((r*(r+1))^2)...
                +(r+2)^2*(r+s+1)*(r-s+1)...
                /((r+1)^2*(2*r+3)*(2*r+1)...
                *(sf-(r+1)*(r+2)));
    else
        M(i) = -(sigma^2*(sf-r*(r+1)))...
                /((r*(r+1))^2)...
                +(r+2)^2*(r+s+1)*(r-s+1)...
                /((r+1)^2*(2*r+3)*(2*r+1)...
                *(sf-(r+1)*(r+2)))...
                +(r-1)^2*(r^2-s^2)...
                /(r^2*(4*r^2-1)*(sf-r*(r-1)));
    end % if
    if (M(i) == inf), M(i) = realmax; end
    end % for
    % build F1 & F2 matix
    f1 = zeros(N2,N2); f2 = zeros(N2,N2);
    for i = 1:N2
    f1(i,i) = M(2*i-1);
    f2(i,i) = M(2*i);
    if (i+1 <= N2)
        f1(i,i+1) = L(2*i-1);
        f1(i+1,i) = L(2*i-1);
        f2(i,i+1) = L(2*i);
        f2(i+1,i) = L(2*i);
    end % if
    end % for
    % symmetric modes
    [v1,d1] = eig(f1); lamb1 = diag(d1);
    [~,ii] = sort(-lamb1);
    lamb1 = lamb1(ii); v1 = v1(:,ii);
    ht1 = 4.d0*a^2*omega^2/g.*lamb1/1000.d0;
    % anti-symmetric modes
```

```matlab
    [v2,d2] = eig(f2); lamb2 = diag(d2);
    [~,ii] = sort(-lamb2);
    lamb2 = lamb2(ii); v2 = v2(:,ii);
    ht2 = 4.d0*a^2*omega^2/g.*lamb2/1000.d0;
 5  % Legendre-Gauss quadrature points
    nlat = 94; [x,w] = lgwt(nlat,-1,1);
    % normalized associated Legendre functions
    prs = pmn_polynomial_value(nlat,N+s,s,x);
    % compute Hough modes
10  h1 = zeros(nlat,N2); h2 = zeros(nlat,N2);
    for i = 1:N2
    for j = 1:N2
    i1 = 2*j+s-1; i2 = 2*j+s;
    for ii = 1:nlat
15  % symmetric modes
    h1(ii,i) = h1(ii,i) + v1(j,i)*prs(ii,i1);
    % anti-symmetric modes
    h2(ii,i) = h2(ii,i) + v2(j,i)*prs(ii,i2);
    end
end
    end
    % put them together
    lamb = zeros(N,1); hough = zeros(nlat,N);
    for i = 1:N2
for j = 1:nlat
    i1 = 2*i-1; i2 = 2*i;
    lamb(i1) = lamb1(i);
    lamb(i2) = lamb2(i);
    hough(j,i1) = h1(j,i);
hough(j,i2) = h2(j,i);
    end
    end
    [~,ii] = sort(1./lamb);
    lamb = lamb(ii); hough = hough(:,ii);
35  % equivalent depth (km)
    h = 4.d0*a^2*omega^2/g.*lamb/1000.d0;
    % compute Hough functions for wind components
    b1 = (sigma^2-x.^2).*sqrt(1.d0-x.^2);
    b2 = sqrt(1.d0-x.^2)./(sigma^2-x.^2);
dhdx = central_diff(hough,x);
    hough_u = diag(s./b1)*hough ...
             - diag(b2.*x./sigma)*dhdx;
    hough_v = diag((s/sigma).*x./b1)*hough ...
             - diag(b2)*dhdx;
```

```
    clf % plot Hough functions
    for j = 1:60
    u = hough(:,j); subplot(10,6,j)
    plot(x, u,'LineWidth',2), grid on
5   end
```

## B2   The Chebyshev collocation method

The second MATLAB code uses the Chebyshev collocation method. It includes a *parity factor* for modes with *odd* zonal wavenumbers ($s$) (Orszag, 1974; Boyd, 1978).

```
    % CHEB_HOUGH - Compute Hough functions
10  % using Chebyshev collocation methods
    clear; format long e
    a = 6.370d6; g = 9.81d0;
    omega = 2.d0*pi/(24.d0*3600.d0);
    %s = 1.d0; sigma = 0.4986348375d0; % DW1
15   s = 1.d0; sigma = 0.5d0;    % DW1
    %s = 2.d0; sigma = 1.0d0;    % SW2
    %s = 3.d0; sigma = 1.5d0;    % TW3
    parity_factor = mod(s,2);
    N = 62; [D1,D2,x] = cheb_boyd(N,parity_factor);
a2 = (1-x.^2)./(sigma^2-x.^2);
    a1 = 2.*x.*(1-sigma^2)./(sigma^2-x.^2).^2;
    a0 = -1./(sigma^2-x.^2).*((s/sigma) ...
        .*(sigma^2+x.^2)./(sigma^2-x.^2) ...
        +s^2./(1-x.^2));
A = diag(a2)*D2 + diag(a1)*D1 + diag(a0);
    [v,d] = eig(A); lamb = real(diag(d));
    % sort eigenvalues and -vectors
    [foo,ii] = sort(-lamb);
    lamb = lamb(ii); hough = real(v(:,ii));
30  % equivalent depth (km)
    h = -4.d0*a^2*omega^2/g./lamb/1000.d0;
    % compute Hough functions for wind components
    b1 = (sigma^2-x.^2).*sqrt(1.d0-x.^2);
    b2 = sqrt(1.d0-x.^2)./(sigma^2-x.^2);
hough_u = diag(s./b1)*hough ...
            - diag(b2.*x./sigma)*D1*hough;
    hough_v = diag((s/sigma).*x./b1)*hough ...
            - diag(b2)*D1*hough;
    clf % plot Hough functions
```

```matlab
for j = 1:60
u = hough(:,j); subplot(10,6,j)
plot(x, u,'LineWidth',2), grid on
end
```

5 And here is the list of the MATLAB codes for computing Chebyshev differential matrices *numerically* with an option for including the parity factor.

```matlab
function [D1, D2, x] = cheb_boyd(N, pf)
% CHEB_BOYD - Compute differential matrix
% for Chebyshev collocation method;
% It contains an optional parity factor (pf)
t = (pi/(2*N)*(1:2:(2*N-1)))';
x = cos(t); n = 0:(N-1);
ss = sin(t); cc = cos(t);
sx = repmat(ss,1,N); cx = repmat(cc,1,N);
nx = repmat(n,N,1);  tx = repmat(t,1,N);
tn = cos(nx.*tx);
if  pf==0
    phi2 = tn;
    PT = -nx.*sin(nx.*tx);
    phiD2 = -PT./sx;
    PTT = -nx.^2.*tn;
    phiDD2 = (sx.*PTT-cx.*PT)./sx.^3;
else
    phi2 = tn.*sx;
    PT = -nx.*sin(nx.*tx).*sx + tn.*cx;
    phiD2 = -PT./sx;
    PTT = -nx.^2.*tn.*sx ...
        - 2*nx.*sin(nx.*tx).*cx - tn.*sx;
    phiDD2 = (sx.*PTT-cx.*PT)./sx.^3;
end
D1 = phiD2 /phi2; % the first derivatives
D2 = phiDD2/phi2; % the second derivatives
```

### Appendix C:  The parity factor for basis functions on the sphere

Orszag (1974), Boyd (1978), Secs. 18.8 and 18.9 of Chapter 18 in Boyd (2001), and Boyd and Yu (2011), all provide a detailed
35 analysis of the "parity factor", $\sin(\varphi)^{\mathrm{mod}(s,2)}$, multiplying each latitudinal basis function. Therefore, we shall content ourselves with a heuristic argument here. Note that the analysis here is restricted to scalars; components of vectors are discussed in Boyd (2001).

If $f(\lambda, \varphi)$ is a smooth (infinitely differentiable) scalar function, then it should be continuous when followed along a meridian over the pole. However, $\lambda$ jumps discontinuously as the poly is crossed. Continuity requires that

$$\lim_{\varphi \to 0} f(\lambda, \varphi) = f(\lambda + \pi, \varphi) \tag{C1}$$

for all $\lambda$. Let us expand in a longitudinal Fourier series

$$f(\lambda, \varphi) = \sum_{s=0}^{\infty} a_s(\varphi) \cos(s\lambda) + b_s(\varphi) \sin(s\lambda) \tag{C2}$$

Because the Fourier basis functions are linearly independent, each term must individually satisfy the continuity condition. All *even* wavenumbers have the property of invariance with respect to translation by $\pi$ and therefore are unchanged when followed along a meridian over a pole:

$$\cos(2s[\lambda + \pi]) = \cos(2s\lambda + 2s\pi) = \cos(2s\lambda), \qquad s = 0, 1, 2, \ldots \tag{C3}$$

However, all *odd* wavenumbers are *sign-reversed*:

$$\cos([2s-1][\lambda + \pi]) = \cos([2s-1]\lambda + [2s-1]\pi) = -\cos([2s-1]\lambda), \qquad s = 1, 2, \ldots \tag{C4}$$

as illustrated in Fig. C.1. The continuity condition cannot be satisfied unless the limit as $\varphi \to 0$ of all Fourier coefficients for all *odd* longitudinal wavenumbers is *the only value that is equal to its own negative, zero*, that is

$$\lim_{\varphi \to 0} a_{2s-1}(\varphi) = 0 \tag{C5}$$

(and similarly for the sine coefficients), as shown schematically in Fig. C.2. The parity factor $\sin(\varphi)$ enforces this zero for all odd wavenumbers. It is unnecessary for even longitudinal wavenumbers because trigonometric functions of even zonal wavenumber are continuous across the poles automatically.

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

**Table 1.** Number of good eigenvalues of three tidal waves DW1, SW2 and TW3 computed with different trunction $N$ using two different methods: I - normalized ALP expansion, II - Chebyshev collocation.

| $N$ | DW1-I | DW1-II | SW2-I | SW2-II | TW3-I | TW3-II |
|---|---|---|---|---|---|---|
| 8 | 2 | 0 | 2 | 0 | 3 | 1 |
| 16 | 6 | 1 | 6 | 5 | 10 | 6 |
| 24 | 10 | 3 | 10 | 9 | 16 | 13 |
| 32 | 16 | 9 | 14 | 13 | 22 | 19 |
| 40 | 22 | 14 | 20 | 18 | 28 | 25 |
| 48 | 28 | 15 | 24 | 22 | 36 | 32 |
| 56 | 32 | 24 | 29 | 27 | 42 | 39 |
| 64 | 38 | 29 | 34 | 32 | 48 | 45 |
| 72 | 43 | 29 | 38 | 37 | 56 | 52 |
| 80 | 49 | 39 | 44 | 42 | 62 | 59 |

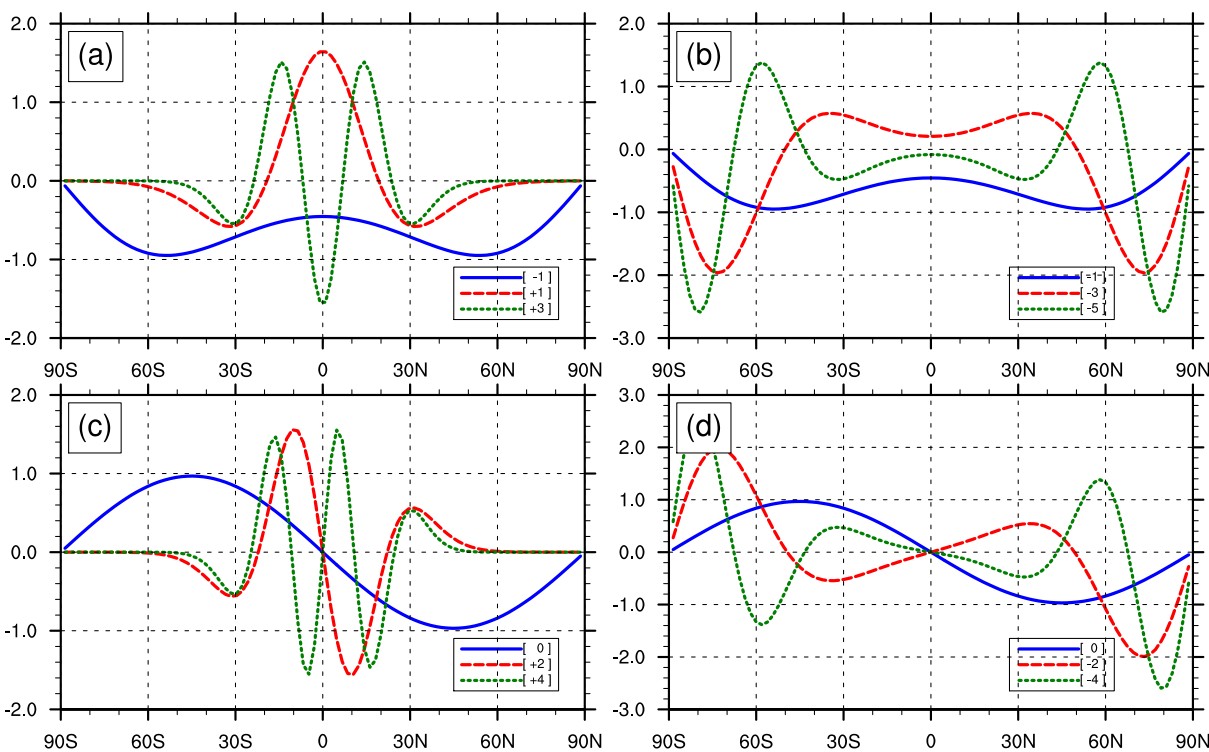

**Figure 1.** The first few symmetric and antisymmetric Hough modes for DW1 ($s = 1, \sigma = 0.5$) of scalar fields, computed using the normalized associated Legendre polynomial (ALP) expansions. Panels (a) and (b) are for symmetric modes, (c) and (d) are for anti-symmetric modes. The labels are: [ -1 ] for the first *negative* mode with largest *negative* eigenvalue, [ +1 ] for the first *positive* mode with largest *positive* eigenvalue, and [ 0 ] for the so-called missing mode with *zero* eigenvalue or *infinite* equivalent depth.

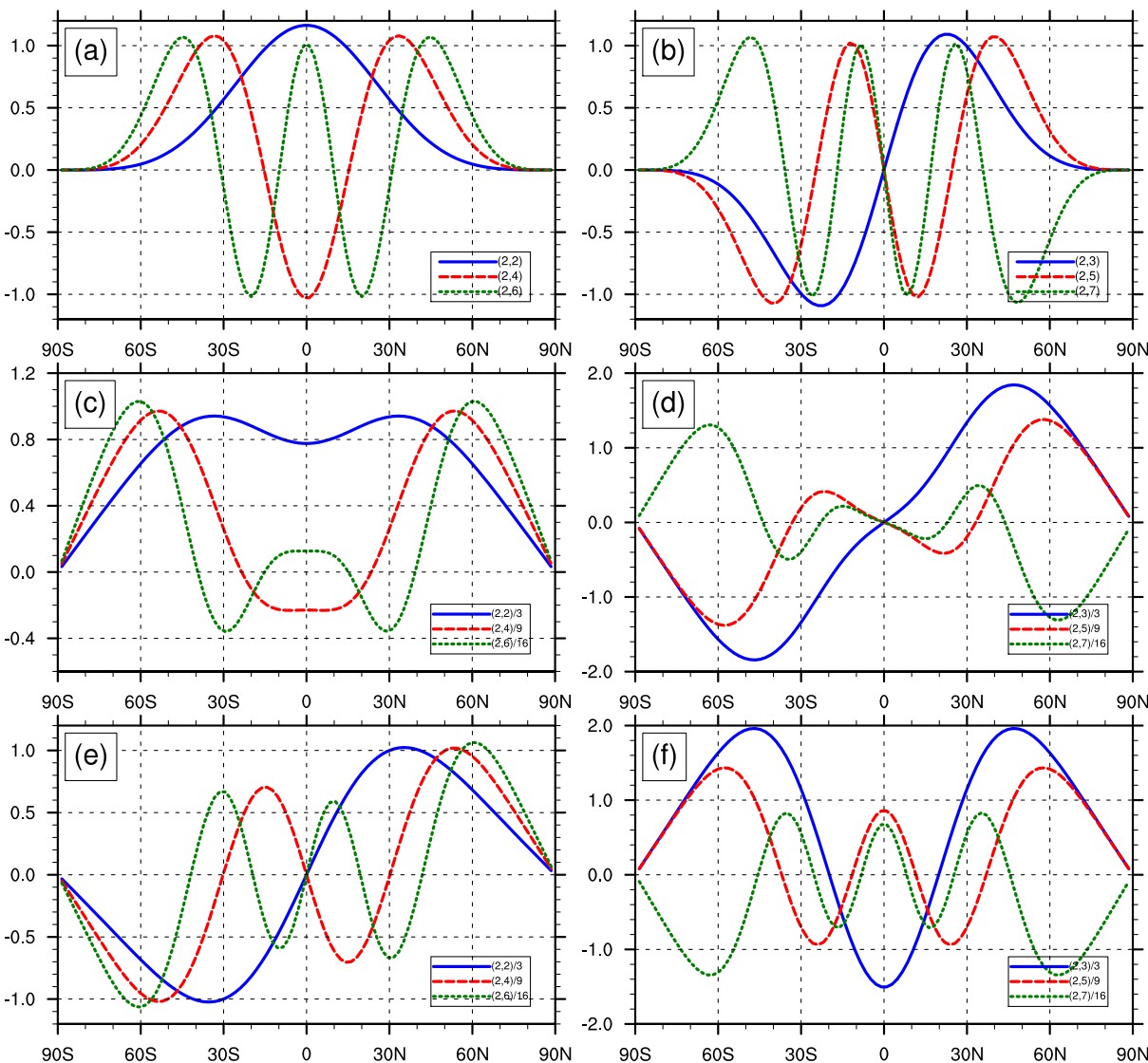

**Figure 2.** The first few symmetric and antisymmetric Hough modes for SW2 ($s = 2, \sigma = 1$), computed using the normalized associated Legendre polynomial (ALP) expansions. The left panels are symmetric modes and the right panels are anti-symmetric modes, except panels (e) and (f) which are reversed. Panels (a) and (b) are for the scalar fields, (c) and (d) for the zonal wind component, (e) and (f) for the meridional wind component. The labels are conventional.

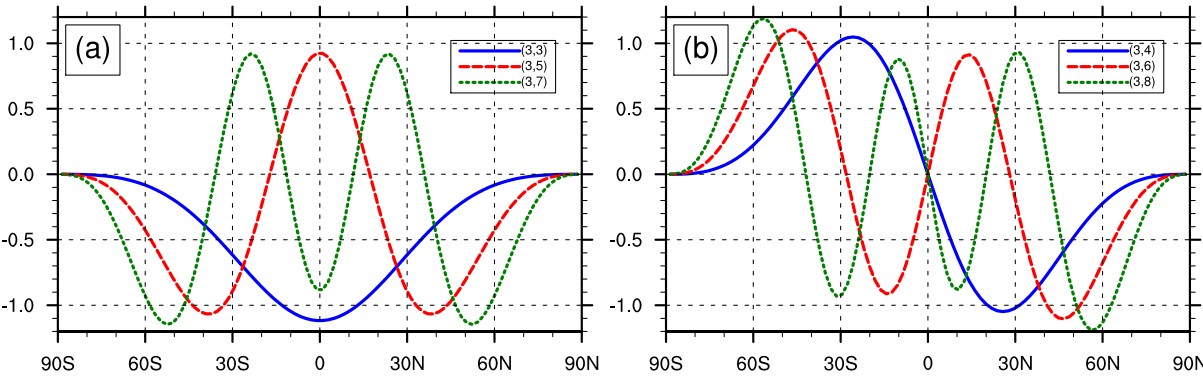

**Figure 3.** The first few symmetric and antisymmetric Hough modes for TW3 ($s = 3, \sigma = 1.5$) of scalar fields, computed using the normalized associated Legendre polynomial (ALP) expansions. The left panels are symmetric modes and the right panels are anti-symmetric modes.

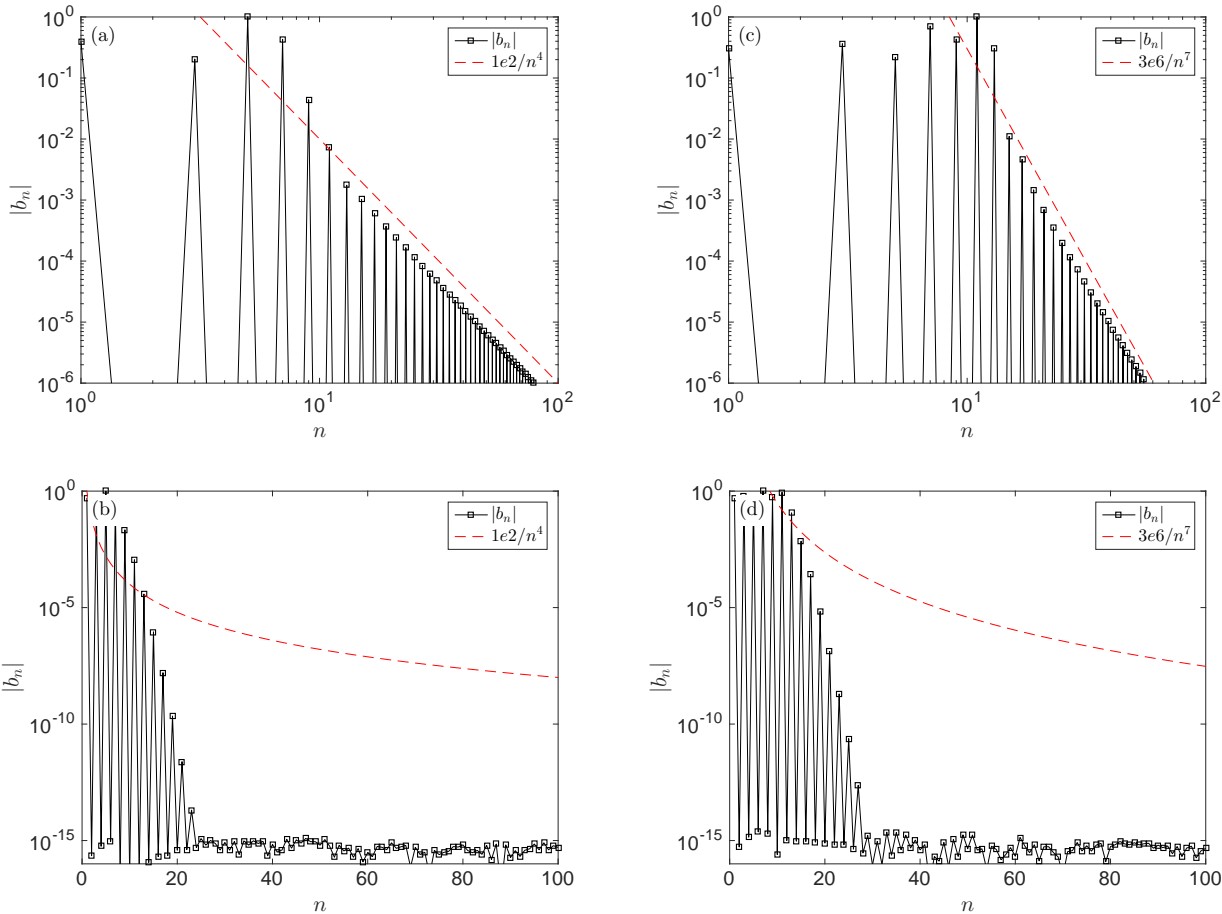

**Figure 4.** The absolute value of the expansion coefficients $b_n$ in Eq. (15), truncated at $N = 150$. The left panels are for the terdiurnal tides, s=3, $\sigma$=1.5, for eigenfunction with eigenvalue $\gamma$=17.2098: (a) without parity factor, (b) with parity factor; The right panels are for penta-diurnal tides s=5, $\sigma$=2.5, for eigenfunction with eigenvalue $\gamma$=22.9721: (c) without parity factor, (d) with parity factor. An empirical fitting curve is also shown in red dash.

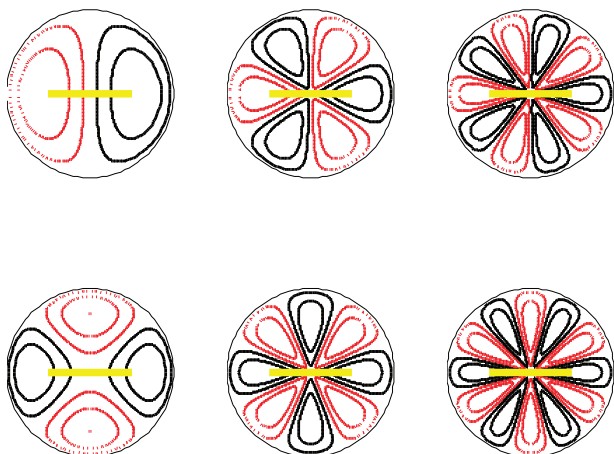

**Figure C.1.** Schematic isolines for Fourier terms $a_s(\varphi)\cos(s\lambda)$ for various zonal wavenumbers $s$, shown in a polar projection. Positive-valued isolines are solid black while negative-valued isolines are red dashed. The thick yellow line segments depict a part of a meridian. For *odd* wavenumbers (upper panels), the yellow lines connect solid black contours to red dashed isolines – the function changes sign along the meridian.

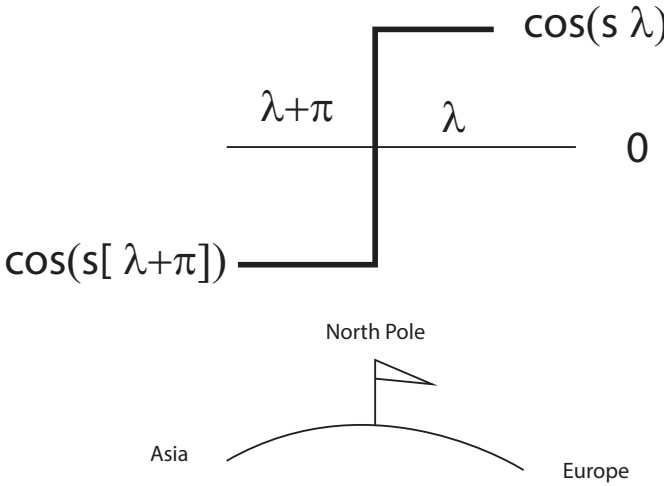

**Figure C.2.** Schematic of the behavior of $a_s(\varphi)\cos(s\lambda)$ along a meridian. If $a_s(0) \neq 0$, the Fourier term will have a jump discontinuity across the pole (thick black curve) when longitude jumps by $\pi$.