# Peer review of "On computation of Hough functions"

_Geoscientific Model Development, 2015_

## Referee Comment (RC1) · Anonymous Referee #1 · 29 Feb 2016

General comments

The authors compare the two methods for computing Hough functions: the one using normalized associated Legendre functions (ALF) and the other using the Chebyshev collocation. I don't see the authors' contributions either on scientific insights on Hough functions or on technical improvements for their computation. The manuscript, however, provides a good review on this subject and MATLAB code provided for the latter method may have educational value. Therefore, I recommend major revisions to elucidate the value of this paper.

Major comments

1. As discussed in the general comments, author's contribution is not clear. What is new from Boyd (1976)? 2. Discuss advantages and disadvantages of Chebyshev method. Your results clearly show that the method using normalized ALF is superior.

What are the problems with the ALF methods? 3. The ALF method lacks the code and the Chebyshev method lacks the details of computation (equations). 4. Comparisons deserve a separate section. Which method is used to compute the reference? I believe the ALF method should be used. How do your results compare with previous studies?

Minor comments

Page 1, Line 7: MATLAB rather than Matlab. Page 2, Line -5: This paragraph is not easy to understand before the equations are shown in the next section. Page 3, Line 1: What is "x = 1"? Page 5, Line 5: I suggest to rewrite the sentence in either forms below. We found that form (6b) rather than (11b) is advantageous . . . It is advantageous to use . . . Note that "advantage" is a transitive verb and requires an object. Form (6b) is chosen to advantage what? Page 7, Line 19: We can use a general-purpose method to solve eigenvalue problem (in the ALF methods). I don't understand why the authors refer the Chebyshev method as general-purpose, implying the ALF methods to be special-purpose or tailored methods.

―――――――――――――――――

---

## Author Comment (AC1) · 14 Mar 2016

Dear Paul,

Our reply to the referee is below. The revised manuscript and the marked-up version are uploaded as supplement.

Thanks for your efforts with our article.

Best wishes,

Houjun Wang

John Boyd

[Figure]

**1 Response to Referee #1**

***General comments*** *The authors compare the two methods for computing Hough functions: the one using normalized associated Legendre functions (ALF) and the other using the Chebyshev collocation. I don't see the authors' contributions either on scientific insights on Hough functions or on technical improvements for their computation. The manuscript, however, provides a good review on this subject and MATLAB code provided for the latter method may have educational value. Therefore, I recommend major revisions to elucidate the value of this paper.*

**Reply:** We thank the referee for his/her constructive comments. We will provide a point-by-point reply below. And we revise, clarify, and expand the manuscript accordingly.

But it may be helpful to state what we think what our article has made explicit and/or elaborated on the following points that can be considered as new and useful contribution to literature on computation of Hough functions:

1. We pointed out a correct way to implement the *normalized* ALF expansion method, which was not explicitly stated in the *limited* previous publications using this method;

2. Although Orszag (1974) stated the importance of including the parity factor for accuracy, but he didn't analyze the rate of convergence when the parity factor was omitted. Therefore, the analysis of convergence rates shown in Fig. 4 of our article represents a new result; and

3. The connection of the symmetric matrices and *completeness* of eigenvalues/eigenvectors are not explicitly stated in the previous publications on Hough functions that we know of. And here it is made *explicit*, even though it may be *obvious* now. But "most research consists mainly in realizing the obvious and that it is a slow and laborious process" (G. K. Batchelor, 1959)

[Figure]

**Major comments**

*1. As discussed in the general comments, author's contribution is not clear. What is new from Boyd (1976)?*

**Reply:** a. Computing speeds have greatly improved. In those days, minimization of floating point operations was the sole criteria of merit. Today, eigenvalues of a $1000 \times 1000$ matrix can be found in half a second on a laptop. For small and medium N where N is the size of the discretization matrix, ease of use and convenience of programming is more important than pure speed.

However, the regime of large N is still interesting for some applications. Our paper compares basis sets on both ease-of-use and floating point speed.

b. Development of fast algorithms for symmetric tridiagonal matrices has altered the efficiency questions we show more clearly in the new draft.

In 1976, most computations were performed on the CDC 6600 which had a floating point speed of 0.6 megaflops when applied to large linear algebra benchmarks. Boyd's allocation of five hours on this machine thus allowed about 10 billion floating point operations. Since the state of the art eigensolver of those times, QR, had a cost of about $O(N^3)$ operations where N is the size of the matrix, Boyd's entire allocation, obtained by writing a short proposal to the NCAR computing program, would have been exhausted by finding the eigenvalues of a single matrix of dimension 1000. However, the CDC 6600 couldn't actually do problems of this size. Its core memory could only store about 50,000 numbers, so a single matrix $200 \times 200$ exhausts memory!

In this environment, efficiency triumphed over other considerations.

In 2016, the question of "what is best" no longer has a unique answer. When the goal is to find thousands of eigenmodes, as might be desirable in Hough function/normal mode analysis of a global weather forecasting model, efficiency matters. The normalized ALF method, which yields a symmetric tridiagonal matrix that can be solved in $O(N^2)$

operations or less versus the $O(N^3)$ required by the dense matrices generated by the Chebyshev method, the normalized ALF method is a clear winner.

However, *in terms of convenience and ease of use, the collocation method using the parity-modified Chebyshev [cosine] series is the clear winner.*

On a modern laptop, $10^{10}$ operations is less than half a second of execution time. Computational speed is now irrelevant for small N.

*2. Discuss advantages and disadvantages of Chebyshev method. Your results clearly show that the method using normalized ALF is superior. What are the problems with the ALF methods?*

**Reply:** Chebyshev polynomials are really just cosines. Much easier to use than ALF. Can be summed and interpolated by the FFT. Recursion is stable. ALF recursion is increasingly unstable as the zonal wavenumber increases, necessitating a bunch of tricks, etc. And the ALF methods are not as easy to program as the Chebyshev methods.

*3. The ALF method lacks the code and the Chebyshev method lacks the details of computation (equations).*

**Reply:** We added the MATLAB code using the normalized ALF method. MATLAB function *pmn_polynomial_value.m* (https://people.sc.fsu.edu/~jburkardt/m_src/ legendre_polynomial/pmn_polynomial_value.m) is used to compute normalized associated Legendre polynomials. MATLAB function *lgwt.m* (http://www.mathworks.com/ matlabcentral/fileexchange/4540-legendre-gauss-quadrature-weights-and-nodes/ content/lgwt.m) is used to compute the Gauss quadrature points. Also considering the cumbersome programming with the normalized ALF method, in computing the Hough functions for horizontal wind components, we use the central difference method with MATLAB function *central_diff.m* (http://www.mathworks.com/matlabcentral/ fileexchange/12-central-diff-m/content/central_diff.m).

We also simplified the portion of the MATLAB code for plotting Hough functions.

Chebyshev method is well described in Boyd's book "Chebyshev and Fourier Spectral Methods" (as referenced in the article). We added the definition of the Chebyshev collocation points and a remark.

***4.a*** *Comparisons deserve a separate section.*

**Reply:** OK, we made subsections out of them.

***4.b*** *Which method is used to compute the reference?*

**Reply:** Doesn't matter as long as the "exact" answer is vey accurate. Both methods are exponentially accurate, so we can use either. We actually used both to check one against the other. We also plot the Chebyshev or ALF coefficients and increase N, the number of degrees of freedom in our benchmarks, until the coefficients reach a "roundoff plateau", in the terminology of Boyd's book, Chapter 2, at around $10^{-13}$.

***4.c*** *I believe the ALF method should be used. How do your results compare with previous studies?*

**Reply:** We agree for large N, but disagree for small N. Also as noted in the article, the advantage of using normalized ALF method, we get symmetric matrices and with all real eigenvalues; and the other methods can get a few inaccurate eigenvalues with nonzero imaginary parts. So an accuracy check, such as by comparing results with different truncations, or with different methods, is always helpful.

***Minor comments***

***Page 1, Line 7:*** *MATLAB rather than Matlab.*

**Reply:** We did a global replacement.

***Page 2, Line -5:*** *This paragraph is not easy to understand before the equations are shown in the next section.*

**Reply:** Move this paragraph to after the equations are shown.

***Page 3, Line 1:*** *What is "x = 1"?*

**Reply:** Changed to $\mu = 1$.

***Page 5, Line 5:*** *I suggest to rewrite the sentence in either forms below. We found that form (6b) rather than (11b) is advantageous ... It is advantageous to use ... Note that "advantage" is a transitive verb and requires an object. Form (6b) is chosen to advantage what?*

**Reply:** Revised to make it more accurate.

***Page 7, Line 19:*** *We can use a general-purpose method to solve eigenvalue problem (in the ALF methods). I don't understand why the authors refer the Chebyshev method as general-purpose, implying the ALF methods to be special-purpose or tailored methods.*

**Reply:** What we mean is that the application of Chebyshev collocation methods doesn't change very much as problems/equations changed: it is usually straightforward to apply the collocation methods to different problems/equations. But for the ALF expansion method, as a Galerkin method, every time the problems/equations changed, such as when the zonal-mean wind is included, the derivations have to be redone again. To quote from Hesthaven et al. (2007, Chapter 3; referenced in our article): "The main drawback of the (Fourier-Galerkin) method is the need to derive and solve a different system of governing ODEs for each problem. This derivation may prove very difficult, and even impossible."

But we removed these statements in case they may cause confusion.

**Supplement:**

[revised manuscript text omitted]

---

## Referee Comment (RC2) · Anonymous Referee #2 · 22 Mar 2016

This paper presents implementation of two numerical methods for computing the eigenvalues and eigenvectors for the Laplace tidal equation, the normalized associated Legendre polynomial expansion and Chebyshev collocation method, which have some advantages over the commonly used unnormalized associated Legendre polynomial expansion method. The authors also show results (Fig 4) that demonstrate how the parity factor in the Chebyshev collocation method affect numerical convergence. A Matlab routine for the Chebyshev method is included in the paper. The implementation is rather straightforward, and the presentation of the paper is clear. I have the following specific comments:

1. Parity factor: It will be helpful if the authors could briefly discuss why the parity factor is dependent on zonal wavenumber.

2. Number of good eigenvalues (page 6 line 21 and Table 1): what are the percentages of good values for these modes using the unnormalized ALP method?

3. Are the computational costs of the two methods comparable? How do they compare with the unnormalized ALP method?

---

## Author Comment (AC2) · 26 Mar 2016

Dear Paul,

Our reply to the referee #2 is below. The revised manuscript and the marked-up version (difference from the revised manuscript in response to referee #1) are uploaded as supplement.

Thanks for your efforts with our article.

Best wishes,

Houjun Wang

John Boyd

[Figure]

**1 Response to Referee #2**

***General comments*** *This paper presents implementation of two numerical methods for computing the eigenvalues and eigenvectors for the Laplace tidal equation, the normalized associated Legendre polynomial expansion and Chebyshev collocation method, which have some advantages over the commonly used unnormalized associated Legendre polynomial expansion method. The authors also show results (Fig 4) that demonstrate how the parity factor in the Chebyshev collocation method affect numerical convergence. A Matlab routine for the Chebyshev method is included in the paper. The implementation is rather straightforward, and the presentation of the paper is clear.*

**Specific comments**

***1.*** *Parity factor: It will be helpful if the authors could briefly discuss why the parity factor is dependent on zonal wavenumber.*

**Reply:** We have added an appendix on the parity factor.

***2.*** *Number of good eigenvalues (page 6 line 21 and Table 1): what are the percentages of good values for these modes using the unnormalized ALP method?*

**Reply:** It turns out that, when both the methods are implemented correctly (and the symmetric and anti-symmetric modes are computed separately, especially for the trickiest DW1 modes), the percentage of good values using the un-normalized ALP expansion method is the same as that of the normalized ALP expansion method. This is understandable as the recursive relationship for the normalized ALP expansion method can be derived directly from the recursive relationship for the un-normalized ALP expansion method.

However, the factorial factors (that convert the un-normalized ALPs to the normalized ALPs) grow rapidly with zonal wavenumber $s$ and latitudinal degree, so we suspect that normalized versus unnormalized differences would appear for larger $s$ and larger Legendre truncations. We can only say that differences are small in the parameter range for atmospherical tidal applications.

*3. Are the computational costs of the two methods comparable? How do they compare with the unnormalized ALP method?*

**Reply:** The computational costs are all very small, about a second or fractions of a second; so for most applications this question is of little concern now (also see our response to Major comment 1 of referee #1).

In addition, we have taken this opportunity to improve and clarify the manuscript in several places, as can be discerned from the marked-up version (difference from the revised manuscript in response to referee #1).

Please also note the supplement to this comment:
http://www.geosci-model-dev-discuss.net/gmd-2015-282/gmd-2015-282-AC2-supplement.pdf

**Supplement:**

[revised manuscript text omitted]